# Assessment of the Sensitivity of the Mean Climate Simulation over West Africa to Planetary Boundary Layer Parameterization Using RegCM5 Regional Climate Model

**Foungnigué Silué** [1,*], **Adama Diawara** [1,2], **Brahima Koné** [1], **Arona Diedhiou** [1,3,4,*], **Adjon Anderson Kouassi** [5], **Benjamin Komenan Kouassi** [1,2], **Fidèle Yoroba** [1,2], **Adama Bamba** [1], **Kouakou Kouadio** [1,2], **Dro Touré Tiémoko** [2,6], **Assi Louis Martial Yapo** [2,7], **Dianicoura Ibrahim Koné** [1] and **Adjoua Moise Landry Famien** [7]

1    Laboratoire des Sciences de la Matière de l'Environnement et de l'Energie Solaire (LASMES), Université Félix Houphouët Boigny (UFHB), Abidjan BP 582, Côte d'Ivoire; adama.diawara23@ufhb.edu.ci (A.D.); kone.brahima3@ufhb.edu.ci (B.K.); komenan.kouassi77@ufhb.edu.ci (B.K.K.); fidele.yoroba49@ufhb.edu.ci (F.Y.); adama.bamba00@ufhb.edu.ci (A.B.); kouakou.kouadio34@ufhb.edu.ci (K.K.); dianikoura_ib@hotmail.com (D.I.K.)

2    Station Géophysique de Lamto, N'Douci BP 31, Côte d'Ivoire; tiemokodro.sfa@univ-na.ci (D.T.T.); martial_yapo@uao.edu.ci (A.L.M.Y.)

3    African Center of Excellence on Climate Change, Biodiversity and Sustainable Agriculture (CEA CCBAD), Université Félix Houphouët Boigny, Abidjan BP 582, Côte d'Ivoire

4    Institute of Environmental Geosciences, Université Grenoble Alpes, IRD, CNRS, Grenoble INP, IGE, 38000 Grenoble, France

5    Laboratoire des Sciences et Technologie de l'Environnement (LSTE), Université Jean Lorougnon Guédé, Daloa BP 150, Côte d'Ivoire; adjonkouassi@ujlg.edu.ci

6    Université Nangui Abrogoua, Abidjan BP 31, Côte d'Ivoire

7    Department of Sciences and Technology, University Alassane Ouattara, Bouaké 01 BPV 108, Côte d'Ivoire; famienmoise@uao.edu.ci

\*    Correspondence: foungnigue.silue@ufhb.edu.ci (F.S.); arona.diedhiou@ird.fr (A.D.)

**Abstract:** This study evaluates the performance of two planetary boundary parameterizations in simulating the mean climate of West Africa using the Regional Climate Model version 5 (RegCM5). These planetary boundary parameterizations are the Holtslag scheme and the University of Washington scheme. Two sets of three one-year simulations were carried out at 25 km horizontal resolution with three different initial conditions. The first set of simulations used the Holtslag scheme (hereafter referred to as Hol), while the second used the University of Washington (UW) scheme (hereafter referred to as UW). The results displayed in this study are an average of the three simulations. During the JJAS rainy season, with respect to GPCP, both models overestimated total rainfall in the orographic regions. The UW experiment represented total rainfall fairly well compared to its counterpart, Hol. Both models reproduced convective rainfall well, with a relatively weak dry bias over the Guinean coast subregion. Globally, UW performed better than Hol in simulating precipitation. The pattern of near-surface temperature in both models was well reproduced with a higher bias with Hol than with UW. Indeed, the UW scheme led to a cooling effect owing to the reduction in eddy heat diffusivity in the lower troposphere contributing to reduce the bias. As a consequence, the height of the planetary boundary layer (PBL) was best simulated using the UW scheme but was underestimated compared to ERA5, while using the Hol scheme failed to capture the height of the PBL. This is coherent with the distribution of total cloud cover, which was better simulated with the UW scheme compared to the Hol scheme. This study shows that use of both planetary boundary parameterizations leads to a good simulation of most of the climatological characteristics of the West African region. Nevertheless, use of the UW scheme contributes to a better performance than use of the Hol scheme, and the differentiation between the two schemes is significant along the Guinea Coast and in orographic regions. In these topographically complex regions, UW appears to be more appropriate than Hol. This study emphasizes the importance of planetary boundary parameterizations for accurately simulating climate variables and for improving climate forecasts and projections in West Africa.

**Keywords:** RegCM5; PBL; rainfall; regional climate model; West Africa

## 1. Introduction

Interactions between the surface, which has a vast and diverse ecosystem, and the atmosphere are predominant in the layers of the atmosphere closest to the Earth's surface, known as the planetary boundary layer (PBL). This interface between the free atmosphere and the surface is the seat of turbulent eddies and important interactions of microscale and mesoscale meteorological phenomena such as vertical fluxes of momentum, heat, and mass [1]. Additionally, the Earth's atmosphere is constantly undergoing a number of constraints from its environment that force it to transform, restructure, and move.

The resulting global atmospheric circulation plays a vital role in the planet's climate, ensuring the transfer of energy between warm intertropical and cold polar regions. Several studies based on numerical simulations using general circulation models (GCMs) and regional climate models (RCMs) have shown that atmospheric circulation and surface climate are largely sensitive to surface–atmosphere interactions [2–7].

Unlike coarse-resolution GCMs, finer-resolution RCMs offer the possibility of simulating climate variations at very high spatial resolutions, which makes them particularly attractive, as they consider land surface heterogeneity and small-scale forcings such as complex topography and surface processes [8–10]. RCMs have been used several times to study the climate of West Africa and many other regions [8–10].

The regional climate model (RegCM, available from the International Center for Theoretical Physics, ITCP) is one such model. Owing to its good performance over different domains around the globe with the exception of polar regions, RegCM is one of the most widely used models [11].

Several sensitivity analyses involving the selection of an appropriate integration domain, adequate horizontal resolution, applied physical schemes, and adaptation tools have been performed using the RegCM model. Such studies have been conducted in Asia [12–14], Europe [15,16], Latin America [17,18], the Middle East [11], the United States [19] and Africa [5,20–26].

Most of these studies on an appropriate region or domain are involved in solving the main challenge of selecting an appropriate set of physics-parameterization schemes. A clear and quantitative understanding and representation of this interaction can affect climatic variables such as near-surface temperature, precipitation, and the vertical distribution of atmospheric water vapor and clouds, thus affecting the reliability of climate forecasts [3,27]. For example, Kang et al. [28] discussed the choice and effects of convective parameterization on the climatology of the East Asian summer monsoon. Overall, they noted that no single scheme performs better than another in all aspects of simulated climatology. Dutta et al. [29] also showed the sensitivities of ice-phase microphysics and convection on the Indian summer monsoon rainfall (ISMR) and the monsoon intraseasonal oscillation (MISO). Bao et al. [30] compared the parameterizations of the Tiedtke and Grell convection schemes in the simulation of the East Asian summer monsoon climate using the RegCM. They found that the Tiedtke scheme was more likely to activate convection in the lower troposphere than the Grell scheme because of the greater amount of moist static energy available to activate and sustain the development of convective systems. Koné et al. [24] used the RegCM4 model coupled to the CLM4.5 surface model to assess the performance and sensitivity of the simulated West African climate system to different convection schemes. They concluded by suggesting that the Emanuel convective scheme had the best performance for simulating the surface climate in Africa.

Afiesimama et al. [21] examined and assessed the mean state and interannual variability of West Africa's climate as simulated by RegCM3. Their analysis showed that the mean and extreme precipitation patterns over the region were well represented by the model. Adjon et al. [25] concluded that the replacement of the BATS ground surface

scheme by CLM4.5 in the RegCM4 model configuration mainly leads to an improvement in precipitation over the Atlantic Ocean, but the impact is not sufficiently perceptible over the continent. Anwar et al. [17] examined the sensitivity of the Amazonian surface climate to two RegCM4 land hydrology schemes, namely the default TOPMODEL (TOP) scheme and the alternative variable infiltration capacity (VIC) scheme. Their results showed that VIC does not improve the simulation quality over TOP, suggesting the need for further calibration of VIC surface parameters using in situ observations of the Amazon.

On the other hand, the same authors, Anwar et al. [31], with the same model found that VIC is better than TOP in simulating the surface climate in tropical Africa.

PBL has a significant influence on surface climate parameters such as temperature and precipitation. For example, the ground can be warmed or cooled by a change in the boundary layer. Stable and unstable boundary layer conditions affect the wind speed [32].

Using the WRF model, Flaounas et al. [33] investigated the sensitivity of the West African monsoon to parameterization of convection and PBL. One of their conclusions was that PBL patterns have a greater impact on the temperature, vertical moisture distribution, and precipitation amount.

In RegCM, there are two PBL parameterizations: that of Holtslag [34], present since the first version, which has undergone several modifications, and the one introduced more recently by the University of Washington [35].

To our knowledge, no studies have been carried out in West Africa concerning the evaluation of PBL parameterizations in RegCM, and very few studies have been carried out in other regions.

Studies on the sensitivity and evaluation of these two PBL schemes are limited. In Central Europe, Güttler et al. [16] conducted an evaluation study of different PBL parameterizations in RegCM4.2. They noted that the differences in the temperature trend due to the PBL schemes are mainly localized in the lower troposphere, with the schemes showing much greater diversity in the way that vertical turbulent mixing of the water vapor mixing ratio is governed.

Using RegCM4.7 at 25 km resolution, Lagare et al. [13] investigated the influence of the two PBL parameterizations, namely Holtslag and UW, on tropical cyclones (TCs) over the Philippine region. Their study revealed that small biases are obtained in the number of TCs detected from both simulations, and only the UW scheme was able to simulate strong TCs (category 4–category 5).

Komkoua Mbienda et al. [36] examined the performance of the same two parameterizations in the RegCM4.6 regional climate model over Central Africa. They noted that the Holtslag scheme is more favorable for simulating precipitation in Central Africa than the UW and that the latter is better for simulating temperature.

The main objective of the present study is to evaluate the performance of two physical PBL parameterizations, i.e., Holtslag and UW, in RegCM5.0 to simulate climate variability over West Africa. More specifically, this study aims to find the appropriate PBL scheme for simulating surface climate in West Africa.

The paper is structured as follows: The description of the model, data and numerical experiments used in this study are described in Section 2; Section 3 analyzes and discusses the model performance under both PBL parameterizations; and the main conclusions are summarized in Section 4.

## 2. Materials and Methods

### 2.1. Model Description

The model used in this study is the latest version of the regional climate model (RegCM5.0), which was developed and improved by the modeling team at the International Center for Theoretical Physics (ICTP) and the Institute for Atmospheric and Climatic Sciences (ISAC) of the Italian National Research Council (CNR). RegCM5 includes both hydrostatic and non-hydrostatic dynamical cores as well as multiple physics options. It can be run over any region of the world as a limited area model [15] or using a tropical band

configuration. The model also runs with finite-difference discretization using a terrain-following σ (sigma) pressure vertical coordinate system and an Arakawa B-grid finite differencing algorithm [15] and aims to study mesoscale processes in the atmosphere over a selected area of the Earth. The main evolution of RegCM5 compared to the previous version of the model is the inclusion of the dynamic core of the non-hydrostatic weather forecasting model [37,38] developed at CNR-ISAC. In addition, a number of improvements to the physics of the model have also been implemented.

Note that the MOLOCH dynamical core is incorporated as an additional option so that the user can now choose between three descriptions of dynamics: hydrostatic RegCM4 [15], non-hydrostatic RegCM4-NH [39], and MOLOCH [38,40]. The model has open-source code and is available for download (different versions) at https://github.com/ictp-esp/RegCM (accessed on 15 June 2023). A more detailed description of RegCM5 available at the website https://zenodo.org/record/7548172#.Y8gVV7TMKUk (accessed on 15 June 2023) was provided in a study by Giorgi et al. [40].

To carry out the various simulations, RegCM5 with MOLOCH coupled to the Community Land Model version 4.5 (CLM4.5) [41] was used. As the main objective is to assess the sensitivity of RegCM5 over West Africa to two boundary layer schemes, we carries out two set of three experiments with different initial conditions.

The first set of simulation used the Holtslag scheme [34] (hereafter referred to as Hol), while the second used the University of Washington (UW) scheme [42] (hereafter referred to as UW). For each configuration, we ran three simulations of one year with three initial conditions (1 January 2002, 1 January 2003, and 1 January 2004). For each simulation, the model integration period started on 1 January and ended on 31 December for every year. The first week of each January (from 1 to 7) was discarded as model spin-up and not included in the analysis. For each configuration, the results shown in this study are an average of the three simulations. The study focuses on the rainy season from June to September (JJAS). The model was integrated over the domain of West Africa depicted in Figure 1 with a 25 km (182 × 114 grid points; from 20° W to 20° E and from 0° to 30° N) resolution and a temporal resolution of 6 h (00:00, 06:00, 12:00, and 18:00 UTC). It employed 18 vertical sigma levels, with a model top at 50 hPa. The domain is large enough to accurately simulate the main climate characteristics.

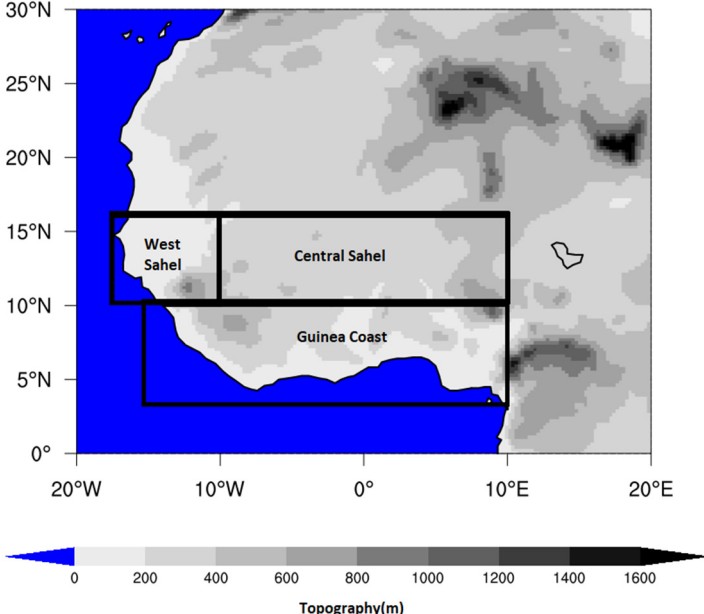

**Figure 1.** Topography of the West African domain. The analysis of the model results focuses on the West African domain and the three subregions, namely Guinea Coast, Central Sahel, and West Sahel, which are marked by black boxes.

The main analysis domain lies between latitudes 0° S–30° N and longitudes 20° W–20° E, with subregions used for more detailed analysis.

Ground surface parameterization as achieved using CLM4.5. Vegetation cover was prescribed and known as satellite phenology (SP) mode [43] from the Moderate Resolution Imaging Spectroradiometer dataset [44]. Also, runoff was handled by the variable infiltration capacity (VIC) scheme [45]. The model physics configuration for the two simulations are summarized in Table 1.

The rapid radiation transfer model [46] as used as the radiation scheme, and the cumulus convection scheme of Emanuel [47] was applied over both land and ocean areas. The mixed-phase microphysics scheme of Hong et al. [48] and the WRF-Single-Moment-MicroPhysics class 5 (WSM5) were used as the microphysics schemes.

**Table 1.** Model configuration.

| Model | Set Up and Simulation Schemes |
|---|---|
| Dynamics | Non-hydrostatic, height-based coordinate MOLOCH [37,38] |
| Horizontal resolution | 25 km × 25 km |
| Vertical levels | 18 |
| Period | 2002–2004 |
| Initial condition | ERA-Interim reanalysis 0.75° × 0.75° resolution at 6-h intervals |
| Lateral boundary condition | OISST (seminal) |
| Microphysics scheme | W5MS [48] |
| Land surface scheme | CLM4.5 [42] |
| Vegetation cover | Satellite phenology (SP) [43] |
| Runoff scheme | Variable infiltration capacity [45] |
| Radiative transfer scheme | RRTM [46] |
| Convection scheme (ocean and land) | Emanuel [47] |
| Ocean flux scheme | Zeng [49] |
| PBL schemes | 1-Holtslag [34] 2-UW [35] |

The simulations were performed using initial and lateral boundary conditions (ICBC), weekly sea surface temperature (SST) fields from the optimal interpolation sea surface temperature (OI_WK), and atmospheric conditions from the 0.75° × 0.75° resolution ERA-Interim reanalysis at 6-h intervals.

As the study evaluates the performance of the two PBL schemes, a brief description of both and the difference between them is presented below.

### 2.1.1. Holtslag PBL Parameterizations

Developed by Holtslag et al. [34], the Holtslag scheme is based on a nonlocal diffusion concept, which means that it uses global mean values to calculate turbulent fluxes within the mixing layer. A particular feature of this scheme is that mixing in the layer is forced only by surface fluxes, that is, heating due to incident solar radiation and friction with the surface, leading to the generation of eddies that are responsible for mixing in the layer. A final important factor concerning the Holtslag scheme is its limited role in the CLP. In the region above this layer, a different approximation is used to represent the physical

processes that occur, where the temperature tendency owing to vertical turbulent mixing is computed in RegCM as follows:

$$\left(\frac{\partial p^* T}{\partial t}\right)_{PBL} = p^* \frac{\partial}{\partial z}\left(K_H\left(\frac{\partial \theta}{\partial z} - \gamma\right)\frac{\Pi}{c_p}\right) \tag{1}$$

where $p^* = p_{SURF} - p_{TOP}$ represents the difference between the surface pressure and pressure at the top of the model. $T$ is the air temperature, $\theta$ is the potential temperature, $K_H$ is the eddy heat diffusivity, and $\gamma$ is a counter-gradient term that parameterizes the dry deep convection transport. $\Pi$ is the Exner function and is the specific heat capacity of dry air at a constant pressure.

In the Holtslag scheme, $K_H$ inside the PBL is determined as $K_H = k w_t z \left(1 - \frac{z}{h}\right)^2$, where $k = 0.4$ is the von Karman constant, $z$ is the height inside the PBL, $w_t$ is the turbulent velocity scale, and $h$ is the PBL height.

### 2.1.2. University of Washington (UW) PBL Parameterizations

The UW scheme [35] is the second PBL scheme in RegCM5. Unlike Holtslag, UW is a local scheme, which means it uses simulation location variants to calculate turbulent flows. In contrast to Holtslag, the UW scheme simulates the entire atmospheric column, i.e., the flows both inside and above the PBL. The UW scheme was developed to address moist thermodynamic processes (i.e., mixing between clear and cloudy air). Its core prognostic equations were written to predict the liquid water potential temperature, total water mixing ratio, and momentum. The model prognostically determines the turbulent kinetic energy (TKE), and it uses TKE to define the diffusivities.

The eddy diffusivity $K_H$ in UW scheme is related to the turbulent kinetic energy (TKE) following Mellor and Yamada [50]:

$$K_H = l \times S_H \sqrt{2TKE} \tag{2}$$

where $l$ is the master turbulent length scale, and $S_H$ is the stability function described in Galperin et al. [51].

The last essential difference between the UW and Holtslag schemes is that layer mixing and turbulence generation are not only forced by surface fluxes. The UW model takes into account the production of turbulence by radiative cooling at cloud tops. In the case of cloud-topped PBL, a term is added to the TKE balance equation. This term is crucial to ensure the production of turbulence in the otherwise stable stratocumulus regions.

### 2.2. Data and Method

The African region faces a lack of high-quality reference databases at an appropriate spatial and temporal resolution. By using different sources of in situ and satellite observation data, the associated uncertainties can be taken into account [52]. In that way, the simulated precipitation is validated using the following observational datasets: Global Precipitation Climatology Project version 1.3 (GPCP V1.3) [53] $1° \times 1°$ resolution products available from 1996 to 2017. Nikulin et al. [52] found a significant dry bias over tropical Africa in TRMM compared to GPCP. However, more recently, Dutta et al. [54] compared GPCP and TRMM data over the global tropics and found a highly spatial correlation. Sylla et al. [55] reported that over Africa, GPCP is more consistent with gauge-based observations.

To validate the simulated 2 m temperature, the fifth-generation reanalysis (i.e., ERA5) from the European Centre for Medium-Range Weather Forecasts (ECMWF) [56] with $0.25° \times 0.25°$ horizontal resolution from 1959 onwards was used. The ERA5 reanalysis is available for download at the electronic address https://cds.climate.copernicus.eu/cdsapp#!/dataset/reanalysis-era5-single-levels?tab=form (accessed on 15 September 2023). The ERA5 products were also used to evaluate the simulated atmospheric fields (total cloud cover and boundary layer height). Dutta et al. [54] showed that the spatial distribution

of total cloud cover from ERA5 climatology is in good agreement with satellite-based CALIPSO-GOCCP.

As the resolution of our study was 25 km, we used the ERA5 product with a higher spatial resolution of approximately 27 km, so the error (due to interpolating the ERA5 data onto the study grid) could be minimized. The other reanalysis products were bilinearly interpolated onto the RegCM5-CLM45 horizontal grid.

All reanalysis products were regridded using bilinear interpolation [52] to the RegCM5-CLM45 horizontal grid to facilitate comparison between the RegCM5 output products and observations. To assess model performance, our study focused on precipitation, air temperature at 2 m, and cloud cover in the summer season from June to September (JJAS).

For quantitative assessment, three statistical tools were used: mean bias (BIAS), root mean square error (RMSE), and model correlation coefficient (PCC). The RMSE and PCC provide information on model performance at the grid point level and are therefore rigorous tests of model performance, whereas the BIAS provides information at the regional or subregional level and is therefore a measure of systematic model errors. The study also focused on four subregions, as in Koné et al. [24]. Each subregion presents different characteristics of the annual cycle of rainfall: Central Sahel (10° W–10° E; 10° W–16° N), West Sahel (18° W–10° W; 10° W–16° N), Guinea Coast (15° W–10° E; 3° W–10° N), and West Africa (20° W–20° E; 5° S–21° N).

## 3. Results

### 3.1. Rainfall

3.1.1. Rainfall Climatology

Rainfall plays a vital role in West Africa and in the Sahel region, where economies, livelihoods, and food security depend heavily on rain-fed agriculture.

In this section, we examine the influence of both Hol and UW configurations on rainfall.

Figure 2 shows the spatial distribution of the mean summer (JJAS) rainfall climatology over West Africa and its associated biases.

Table 2 reports the three specific quantitative performance measures: the MB and PCC between the simulated and observed total rainfall calculated for the entire West African region and for the Gulf of Guinea and the two Sahel subregions.

**Table 2.** Mean bias (MB) for JJAS precipitation for Hol and UW with respect to ERA5 over West Africa and the subregions Guinea Coast, Central Sahel, and West Sahel and PCC for only West Africa.

| | Guinea Coast | Central Sahel | West Sahel | West Africa | |
|---|---|---|---|---|---|
| | MB (%) | MB (%) | MB (%) | MB (%) | PCC |
| Hol | −18.37 | 92.91 | −15.97 | 102.28 | 0.596 |
| UW | −28.81 | 13.23 | −69.65 | −3.47 | 0.778 |

The GPCP total rainfall climatology (Figure 2a) shows the Intertropical Convergence Zone (ITCZ) within a zonal band extending from the Guinean Highlands to the Gulf of Guinea. In this zonal distribution, rainfall intensity decreases from south to north, with maxima in the orographic regions over the Guinean highlands (GH), Jos Plateau (JP) in Nigeria, and Cameroon Mountains (CM). These general patterns of GPCP total precipitation were relatively well simulated by both configurations, namely Hol (Figure 2b) and UW (Figure 2c), with PCC values of 0.57 and 0.77, respectively (Table 2). However, the total precipitation bias varied considerably from Hol to UW in terms of magnitude, sign, and spatial spread. Figure 2d,e show the spatial distribution of the total precipitation biases with respect to the GPCP over almost the entire studied domain. Hol generally overestimated orographic precipitation around the Jos plateau in Nigeria, the Cameroon Highlands, the northern Chad region (9°–17° N, 13°–20° E), and, to a large extent, the zonal band between latitudes (15°–20° N, 10° W–12° E). Consequently, Hol yielded higher MB values of 92.91% and 102.2%, respectively, over Central Sahel and West Africa as a whole (Table 2). The wet

bias was the dominant feature in the simulated Hol precipitation. Dry biases occurred in the GH and slightly along the Guinean coast and West Sahel, with negative MB values of approximately 18% and 15%, respectively. Compared to Hol, UW showed less intense and less extensive wet biases confined to the Jos Plateau in Nigeria, the Cameroon Highlands, part of the Congo Basin, and part of Central Sahel (Figure 2e) and an MB value around 13% (Table 2).

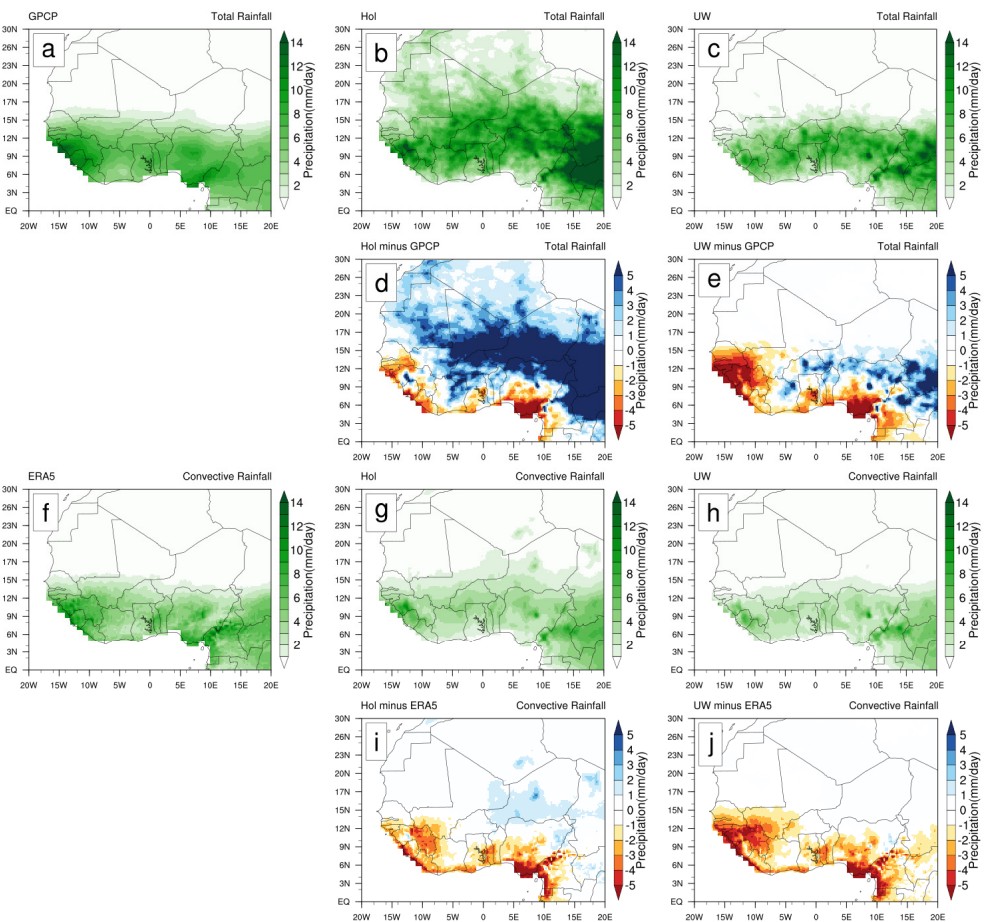

**Figure 2.** Summer mean (JJAS) total rainfall (mm day$^{-1}$) from (**a**) GPCP, (**b**) Hol, and (**c**) UW and convective rainfall (mm day$^{-1}$) from (**f**) ERA5, (**g**) Hol, and (**h**) UW over West Africa and their associated biases: (**d**) Hol minus GPCP, (**e**) UW minus GPCP, (**i**) Hol minus ERA5, and (**j**) UW minus ERA5.

UW dry biases were also observed slightly along the Guinean coast from GH to Cameroon and over West Sahel but were much more pronounced and extended than those of Hol. With MB negative values of approximately 28% and 69%, respectively, over Guinea Coast and West Sahel, UW tended to underestimate globally over the whole West African domain, with a negative MB value of approximately 3%. The total precipitation includes convective precipitation and all other rainfall, mainly from orography. With respect to ERA5, the convective rainfall biases over the entire domain are illustrated in Figure 2 for both Hol (Figure 2i) and UW (Figure 2j). The notable difference between the convective and total rainfall biases is that the dominant wet biases observed with total rainfall are practically non-existent. Therefore, the wet biases observed with the total rainfall are not attributable to convective rainfall. The overestimation of total rainfall is due to other rainfall types, such as those caused by orography on the Jos Plateau in Nigeria and the high plateaus of Cameroon. Only the dry biases remained at similar locations, with high negative MB values for UW at approximately 46% and 68% (compared to 45% and 30% for Hol), respectively, on Guinea Coast and West Sahel. These dry biases were also higher and

more extensive in the UW than in the Hol. The underestimation of total rainfall, mainly along the Guinean coast, seems to be attributable to convective precipitation.

Overall, compared to Hol, UW tended to reduce wet bias in mountainous regions (similar results were shown by Kalmar et al. [57] and Güttler et al. [16]) and increase dry bias in coastal regions.

As noted by Guttler et al. [16], coastal and mountain regions are the most prone to systematic precipitation errors in both models, suggesting perhaps that it may be necessary to use higher horizontal resolutions to reduce the errors associated with the insufficiently resolved orographic phenomenon of increasing precipitation. Thus, a poor simulation of orographic forced ascent or the large uncertainty in the model precipitation estimates over this region [8]. This indicates that both models substantially differ in their ability to simulate the interactions between the WAM elements and deep convection [58] ad suggests that the differences between the RCMs mainly arise from their internal dynamics and physics.

The dominant wet biases observed in the Hol non-local scheme may be due to an overestimation of vertical water vapor transport or to a too-deep boundary layer. The dry biases observed in UW may be due to increased turbulence activity in a region associated with buoyancy perturbations due to clouds intercepting solar radiation.

3.1.2. Rainfall Annual Cycle

In this section, we examine the impact of the PBL configuration on the characterization of the three distinct phases of the West African monsoon (WAM) following the movement of the Inter-Tropical Convergence Zone (ITCZ): the pre-onset (establishment phase), the onset (the period of heavy rainfall), and the southward retreat of the monsoon rain band [59].

A meridional cross-section (time–latitude Hovmöller diagram) is the perfect way to characterize such behavior. This diagram has been widely used to assess the ability of RCMs to simulate seasonal and intraseasonal variations in the West African monsoon and hence the precipitation mechanisms in the region [8,24,60].

Figure 3 shows a Hovmöller diagram of the monthly total and convective rainfall (mm day averaged between 10° W and 10° E and for the period 2002–2004 for GPCP, including (a) observations and each of the models: Hol (b) and UW (c). GPCP total precipitation describes the first phase, i.e., from late April or early May to late June, when spring rains intensify and extend from the Guinean coast to around 5° N. Then, in late June or early July, maximum rainfall moves south of the Sahel, near 12° N, often in the course of a few days [61]. This abrupt change in the latitude of maximum rainfall from the Guinean coast to the Sahel is known as the West African monsoon jump. This marks the start of the rainy season in the Sahel region, with a peak reached in August between 9° N and 12° N, and is accompanied by an abrupt halt in appreciable rainfall intensities along the Guinean coast. In late August, a gradual retreat of the rain band towards the Guinean coast is accompanied by a decrease in rainfall intensity over the Sahel (Figure 3a).

Both Hol and UW models reproduced relatively well the three distinct phases of the WAM annual cycle (Figure 3b,c). Notable differences between models and observations and among models are related to the magnitude and spatial extent of these features. Most of the anomalies mainly concern the peak of monsoon rainfall over the Sahel. For example, the August Sahel monsoon rainfall peak in Hol was more extensive (beyond 15° N) and more intense than that of GPCP and UW. The origin of the discrepancies in the annual cycles of RCMs are mainly due to their different skills in simulating the main features responsible for inducing and maintaining WAM precipitation. Such features include the monsoon flow, the East African Jet (AEJ), the Tropical East Jet (TEJ), and the East African Waves (AEW) [62,63].

Both convective and other rainfall from ERA5 share the same evolution of the total rainfall annual cycle but with different magnitudes and intensities (Figure 3d). With respect to ERA5, both models well reproduced the three WAM phases of convective rainfall, with attenuated intensity and magnitude (Figure 3e,f).

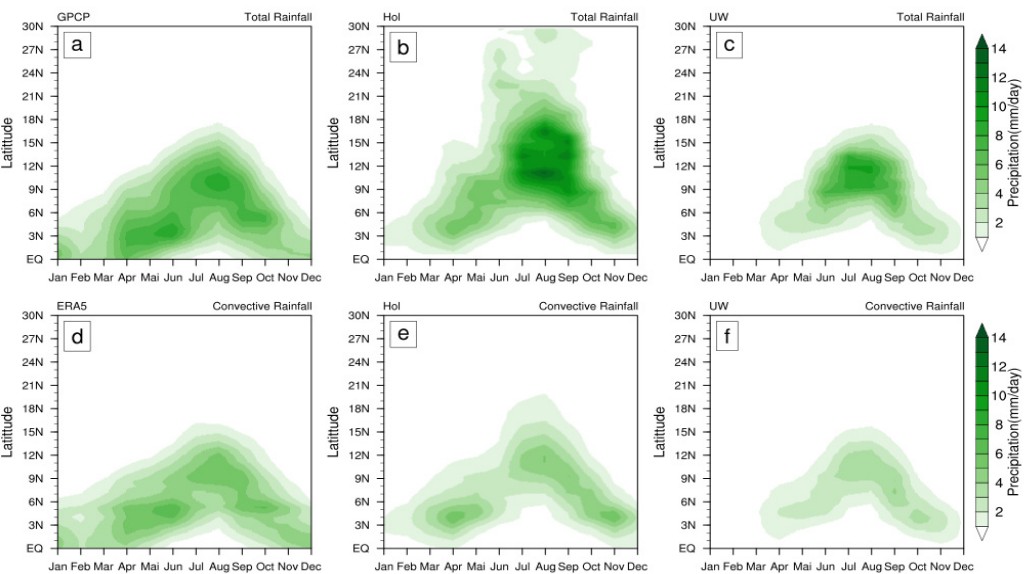

**Figure 3.** Hovmöller diagram of monthly total rainfall (**a**–**c**) and convective rainfall (**d**–**f**) averaged between 10° W and 10° E for observations (GPCP and ERA5) and the models Hol and UW.

An additional analysis of the annual cycle involved zonal averaging of monthly precipitation and temperature climatology across West Africa and the three subregions of Guinea Coast and Central and West Sahel (Figure 4). The aim was to better identify the minima and peaks of precipitation and temperature and, consequently, to better assess the ability of both the Hol and UW models to capture the phases and amplitudes of these variables over the course of the year and in relatively homogeneous regions.

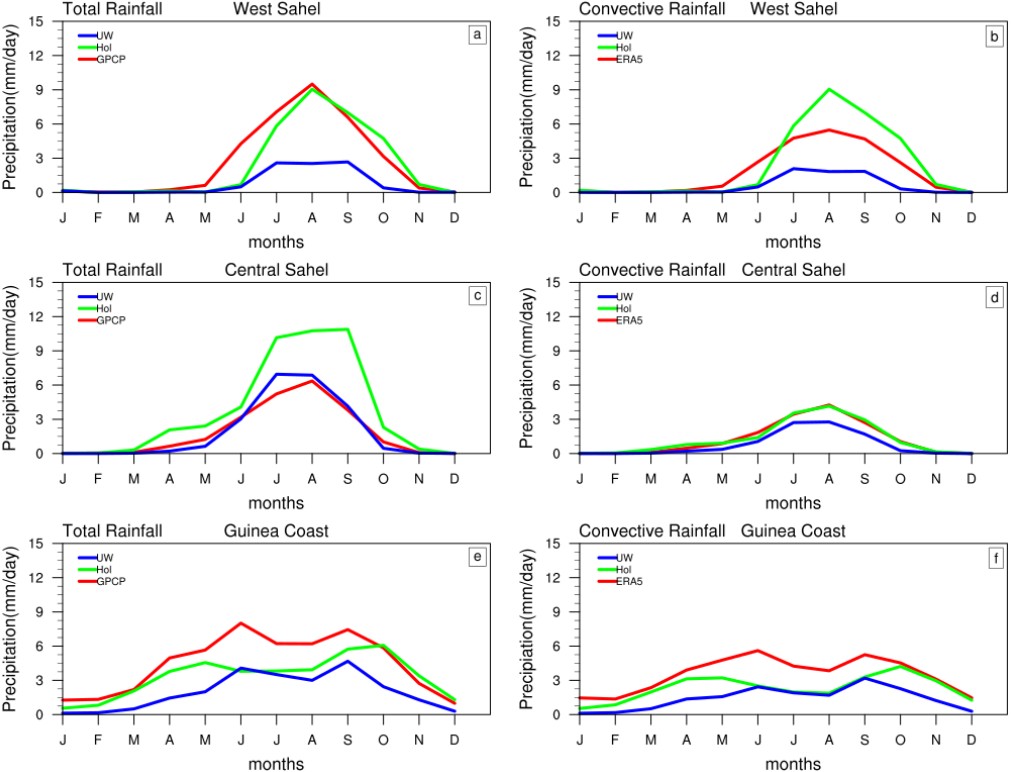

**Figure 4.** Annual cycle of the monthly total (**a**,**c**,**e**) and convective (**b**,**d**,**f**) rainfall (mm day$^{-1}$) averaged over West Sahel, Central Sahel, and Guinea Coast for the observations GPCP and ERA5 and the models Hol and UW.

Figure 4 shows the annual cycle of monthly total and convective rainfall (mm day$^{-1}$) averaged over West Sahel, Central Sahel, and Guinea Coast for the observations GPCP and ERA5 and the models Hol and UW. In general, the profiles underline the progressive meridional evolution of seasonal precipitation.

Over Guinea Coast, total rainfall from the GPCP and both models Hol and UW (Figure 4e) displays a bimodal rain cycle, with a clear primary peak in June and a second peak in September–October as the rain belt retreats southwards. The local minimum in August between the two peaks marks the so-called "little dry season" [64]. The main difference between both models and GPCP relates to magnitude. Both models showed lower intensities than the GPCP, and Hol showed a higher intensity than UW.

Convective rainfall profiles from ERA5 and models in the same coastal region are similar to those of total rainfall, only with slightly attenuated magnitudes (Figure 4f). Convective rainfall accounts for the largest part of the total rainfall phases and magnitudes. This is consistent with the fact that highly organized mesoscale convective systems (MCS) are the main rain-bearing systems in southern West Africa [65].

In contrast, in both Central Sahel and West Sahel, total rainfall from GPCP and models displayed a unimodal profile that peaks in August. This generally corresponds to rainfall data annual profiles in the respective regions [25,66]. The differences are in magnitudes. UW properly equaled the observed magnitude of the GPCP peak in Central Sahel and strongly underestimated it in West Sahel. Conversely, Hol failed to match the magnitude of the peak in Central Sahel, strongly overestimating it, while succeeding in West Sahel.

Convective rainfall products from ERA5 and the models exhibited a similar unimodal profile to those of total rainfall in Central Sahel and West Sahel. Convective precipitation products in Central Sahel share the same magnitude, which is well below those of total rainfall (Figure 4e,f). Overall, the fundamental differences between the subregions with regard to the annual cycle were reasonably well addressed by both Hol and UW models.

In summary, the climatology and annual precipitation cycle show that the UW, with a lower MB and a higher PCC over the Gulf of Guinea, the Sahel, and West Africa as a whole, outperforms Hol in precipitation simulation.

*3.2. Temperature*

Temperature Climatology

Figure 5 shows the distribution of mean summer surface temperature (JJAS) over West Africa for the baseline (a) ERA5 and both models, (b) Hol and (c) UW, as well as the different biases with respect to ERA5 (d,e). Table 3 reports the two specific quantitative performance measures, i.e., the RMSE and the PCC between the simulated and observed temperature calculated for the entire West African region and for the Gulf of Guinea and Sahel subregions.

**Table 3.** RMSE and PCC for JJAS precipitation for Hol and UW with respect to ERA5 over West Africa and the subregions Guinea coast, Central Sahel, and West Sahel and West Africa.

| | Guinea Coast | | Central Sahel | | West Sahel | | West Africa | |
|---|---|---|---|---|---|---|---|---|
| | RMSE | PCC | RMSE | PCC | RMSE | PCC | RMSE | PCC |
| Hol | 2.78 | 0.82 | 1.59 | 0.734 | 1.07 | 0.962 | 1.89 | 0.89 |
| UW | 1.52 | 0.87 | 1.63 | 0.878 | 1.96 | 0.958 | 1.39 | 0.94 |

The ERA5 baseline and models Hol and UW are all consistent in describing a zonal configuration with low temperatures (<30 °C) below 15° N and along the Gulf of Guinea and high temperatures (>30 °C) over the Sahara Desert above 15° N. In other words, temperature minima are located above the orographic peaks of the Guinea, Jos, and Cameroon Mountains and maxima around the Sahara Heat Low (SHL) centered at 25° N.

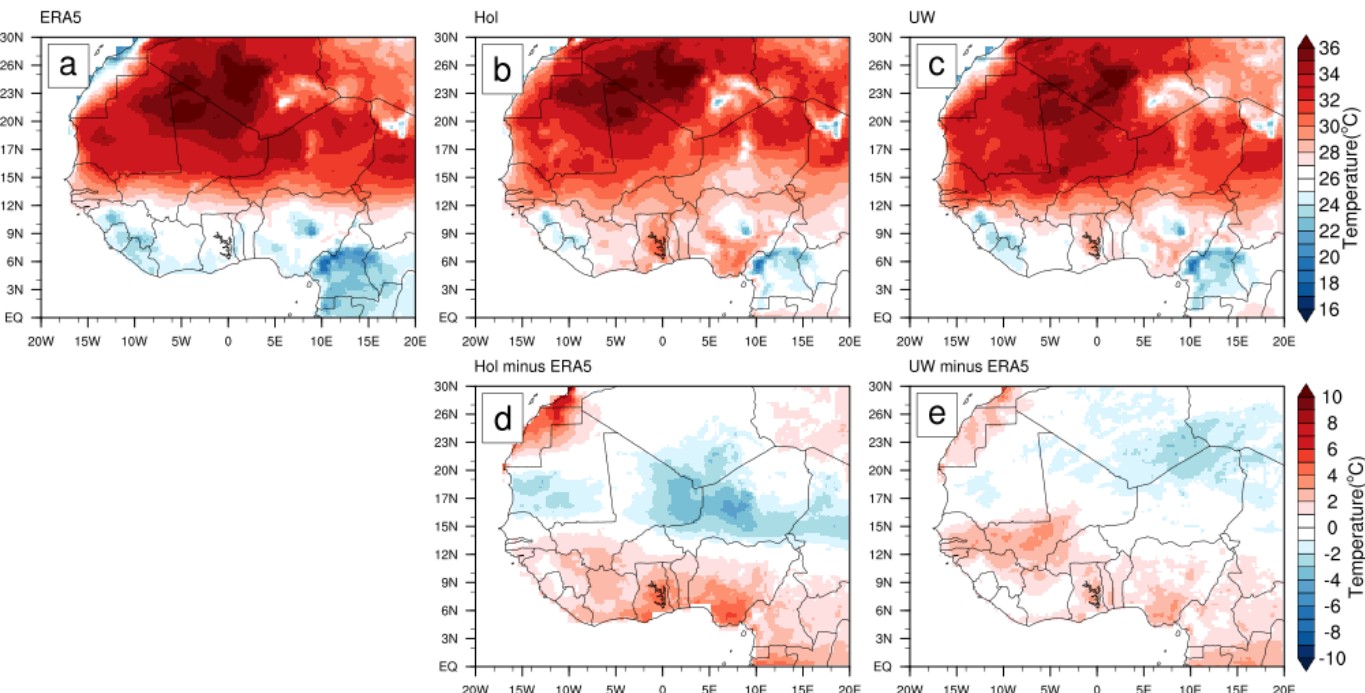

**Figure 5.** Mean summer (JJAS) 2 m temperature over West Africa: (**a**) ERA5 and (**b**) Hol and (**c**) UW and different biases with respect to ERA5 (**d**,**e**).

In summary, with respect to ERA5, the Hol and UW experiments reasonably represented the ground temperatures as well as the South–North temperature gradient with a PCC value around 0.84 and 0.94 and RMSE value 1.89 and 1.39, respectively, over West Africa (Table 3).

This meridional surface temperature gradient remains very important in the formation of the East African Jet [67].

Both models overestimated temperatures along the Guinean coast, the Jos Plateau, the Cameroon Highlands and part of the Congo Basin and inland up to around $10°$ N, with a dominant positive bias of around 2 °C. Then, they presented an underestimation around $25°$ N, with a negative bias of around 2 °C (Figure 5d,e). Nevertheless, both positive and negative biases were more intense and more extensive in Hol than in UW.

This could be related to the fact that the UW scheme increases the cloud cover and thus reduces net surface shortwave flux, resulting in a decrease of the near-surface temperature errors [16].

Below, we examine whether the same conclusion applies to annual cycles. Figure 6 shows the time–latitude temperature diagram following the position and shift of the SHL, which is a major climatological feature during the West African summer monsoon. For example [68] suggested that the monsoon jump is associated with abrupt displacement of the SHL during the summer.

All products, reference ERA5, and both models Hol and UW show a close concordance in reproducing the intensification and northward migration of the SHL from the northern Sahel in April–May to the Sahara in July–August. This migration may initiate a progressive increase in the lower-level temperature gradient between the Gulf of Guinea and the Sahara, reinforcing and moving northwards the phenomena that both induce and maintain WAM. The result is intense convection over the Sahel [3].

Compared to ERA5 and UW, Hol tended overestimate by more than 2 °C the peak of the monsoon period between July and August over the Sahel. This warm bias may produce a stronger temperature gradient that interacts with monsoon features, leading to a more extended and intense band of precipitation. It could explain the higher and more extended overestimation of precipitation in this region (Figure 2b).

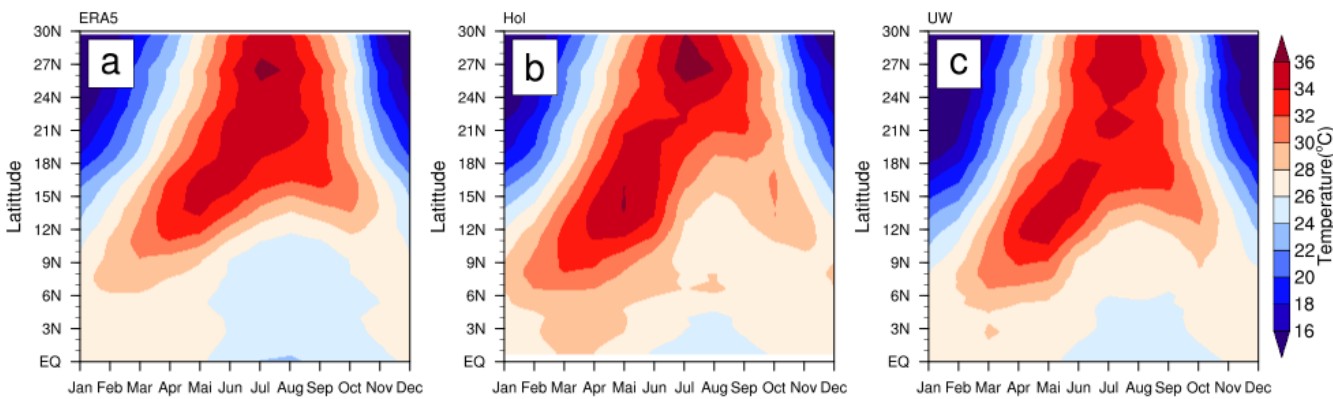

**Figure 6.** Hovmöller diagram of monthly temperature (**a–c**) averaged between 10° W and 10° E for (**a**) ERA5, (**b**) Hol, (**c**) UW.

An additional analysis of the annual cycle involved zonal averaging of monthly precipitation and temperature climatology across the three West African subregions: Guinea Coast, Central, and West Sahel (Figure 7).

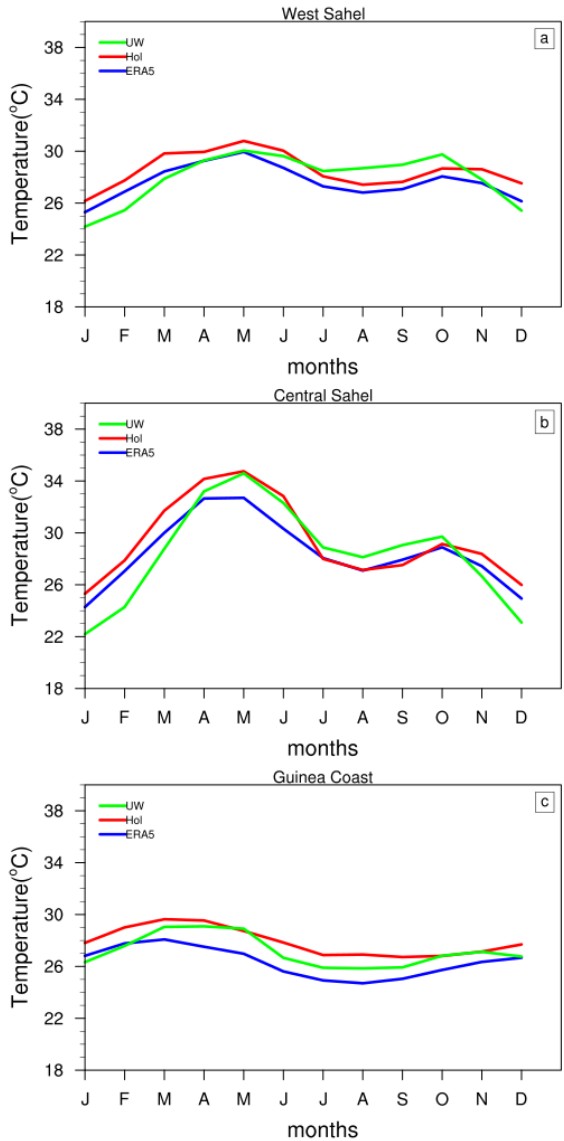

**Figure 7.** Annual cycle of temperature averaged over (**a**) West Central Sahel, (**b**) West Sahel, and (**c**) Guinea Coast for ERA5 and the models Hol and UW.

The annual cycles of temperature for the three subregions (Figure 7a–c) indicate cooler winter and warmer pre- and post-monsoon periods, with relative minima occurring during July–August. Both Hol and UW reproduced the ERA5-observed phase and amplitude well, with a relative error within the range of 2 °C.

The above analysis of the two experiments shows that UW obtained better results than Hol in simulating the near-surface temperature. These results are consistent with those of Komkoua [69] in Central Africa.

The origin of temperature and precipitation biases is difficult to determine accurately, as they depend on many factors, including surface albedo, cloud cover, temperature advection, surface water, energy fluxes, and dust and aerosols [70,71]. Cloudiness as a key factor in the distribution of these quantities is explored in the subsequent sections by analyzing different types of cloud fraction.

### 3.3. Cloud Cover

According to Güttler [16], cloudiness is one of the main factors behind bias in temperature and precipitation.

In this section, we discuss cloud sensitivity in the Hol and UW models and the resulting influence on precipitation and temperature. Figure 8 shows the mean summer (JJAS) total cloud cover (TCC) over West Africa for (a) ERA5, (b) Hol, and (c) UW and different biases with respect to ERA5.

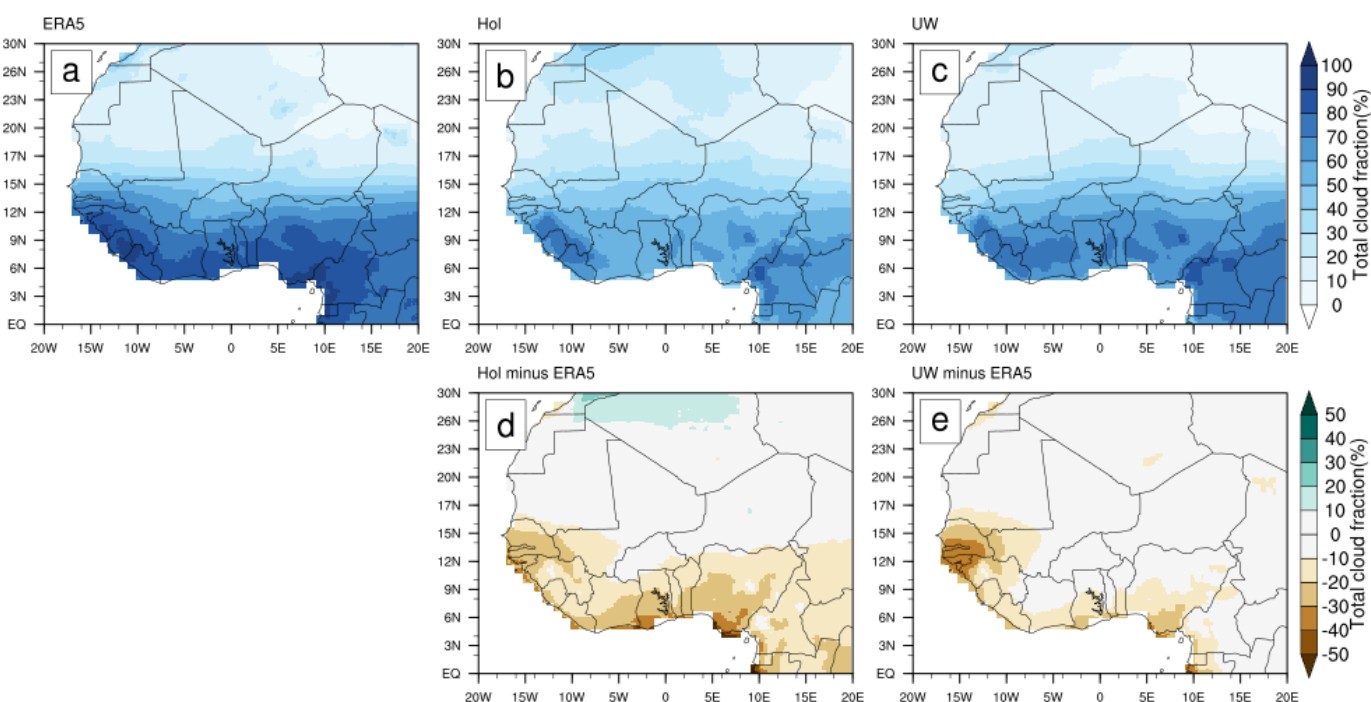

**Figure 8.** Mean summer (JJAS) of total cloud cover over West Africa: (**a**) ERA5, (**b**) Hol, and (**c**) UW and different biases with respect to ERA5.

The spatial distribution of TCC from ERA5 shows strong cloud cover along Guinea Coast and below 15° N and weak cloud cover over the Sahel region above 15° N (Figure 8a).

The Hol and UW configurations reproduced, with remarkable differences in intensity, heavy cloud cover along the Guinean coast and over orographic regions such as the Fouta Jallon Mountains (FJM), Cameroon Highlands (CH), and Jos Plateau (JP) in Nigeria as well as weak cloud cover over the Sahel. Nevertheless, both models reproduced well the distribution of cloud cover in the Sahel region, where PCC values are above 0.8 (Table 4). Both models exhibited poorer TCC patterns (PCC between 0.3 and 0.4) associated with large errors (RMSE > 15) and strong negative biases along the Guinean coast. The negative

biases were higher and more extensive in Hol (around −26%) than in UW (around −16%) (Table 4).

**Table 4.** RMSE, MB, and PCC between ERA5 TCC and simulated TCC from Hol and UW over the entire West Africa and the related three subregions.

|  | Guinea Coast | | | Central Sahel | | | West Sahel | | | West Africa | | |
|---|---|---|---|---|---|---|---|---|---|---|---|---|
|  | RMSE | MB | PCC | RMSE | MB | PCC | RMSE | MB | PCC | RMSE | MB | PCC |
| Hol | 23.30 | −26.51 | 0.47 | 11.41 | 16.27 | 0.89 | 25.42 | 35.08 | 0.89 | 13.46 | −10.16 | 0.95 |
| UW | 15.11 | −16.54 | 0.32 | 6.75 | −5.86 | 0.87 | 1.96 | −38.58 | 0.85 | 1.39 | −9.86 | 0.96 |

The poor performance along the Guinean coast corresponds to an overestimation of the temperature in the same locality.

Clouds, therefore, seem to be one of the factors explaining the temperature bias. As reported by Kalmar et al. [57], the decrease in cloud cover leads to an increase in incident radiation, inducing a higher sensible heat flux and warmer surface temperatures. Therefore, the good performance of UW compared to Hol in the surface temperature simulation is consistent with that of TCC.

In all three subregions, cloud cover for both ERA5 and models Hol and UW was high during the rainy season (June to September) (Figure 9). Compared with ERA5, both models Hol and UW underestimated the intensity of cloud cover. However, this underestimation was much more significant in Hol.

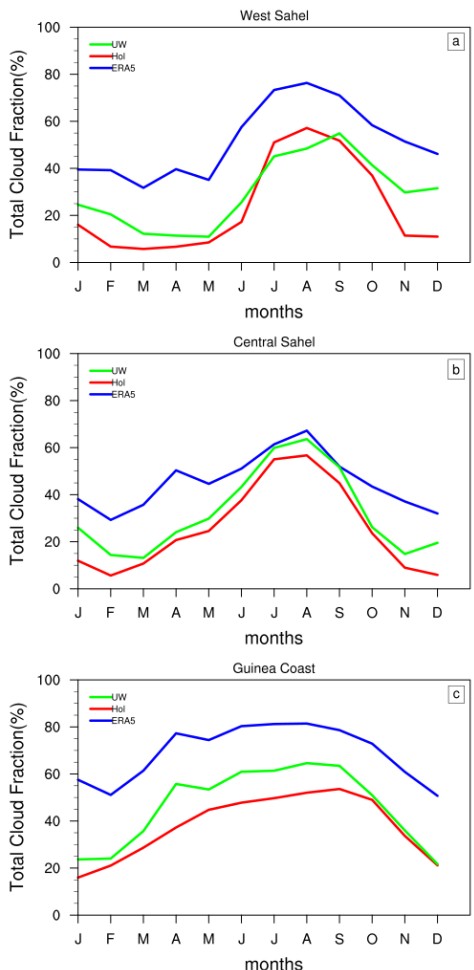

**Figure 9.** Annual cycle of monthly total cloud fraction averaged over West Africa and the subregions (**a**) West Sahel, (**b**) Central Sahel and (**c**) Guinea CoastWest for ERA5 and models Hol and UW.

The decrease in the mean TCC from the south (Guinean coast) to the north of West Africa seems to be controlled by conditions that are essential for cloud formation, such as water vapor and its condensation. Zhang et al. [72] confirmed this hypothesis in their study of cloudiness variations over the Qinghai-Tibet Plateau.

Indeed, under the influence of the monsoon, a greater quantity of water vapor is transported from the ocean and forced to rise by complex topography such as the orographic regions (Fouta Jallon Mountains (FJM), Cameroon Highlands (CH), and Jos Plateau (JP) in Nigeria), favoring conditions for the condensation of moisture into droplets and causing more clouds along the Guinean coast. When the monsoon weakens, and the topography acts as a block, less water vapor reaches the northern part, causing fewer clouds in the hinterland (Sahel).

In summary, UW was closer to ERA5 than Hol in terms of seasonal cycle phases and intensity as well as pattern distribution.

### 3.4. Boundary Layer Height

ERA5 shows high boundary layer height values over West Africa in the dry region over the Saharan Air Layer (SAL) and low values along the coastal region and inland up to 15° N, including orographic regions (Figure 10a).

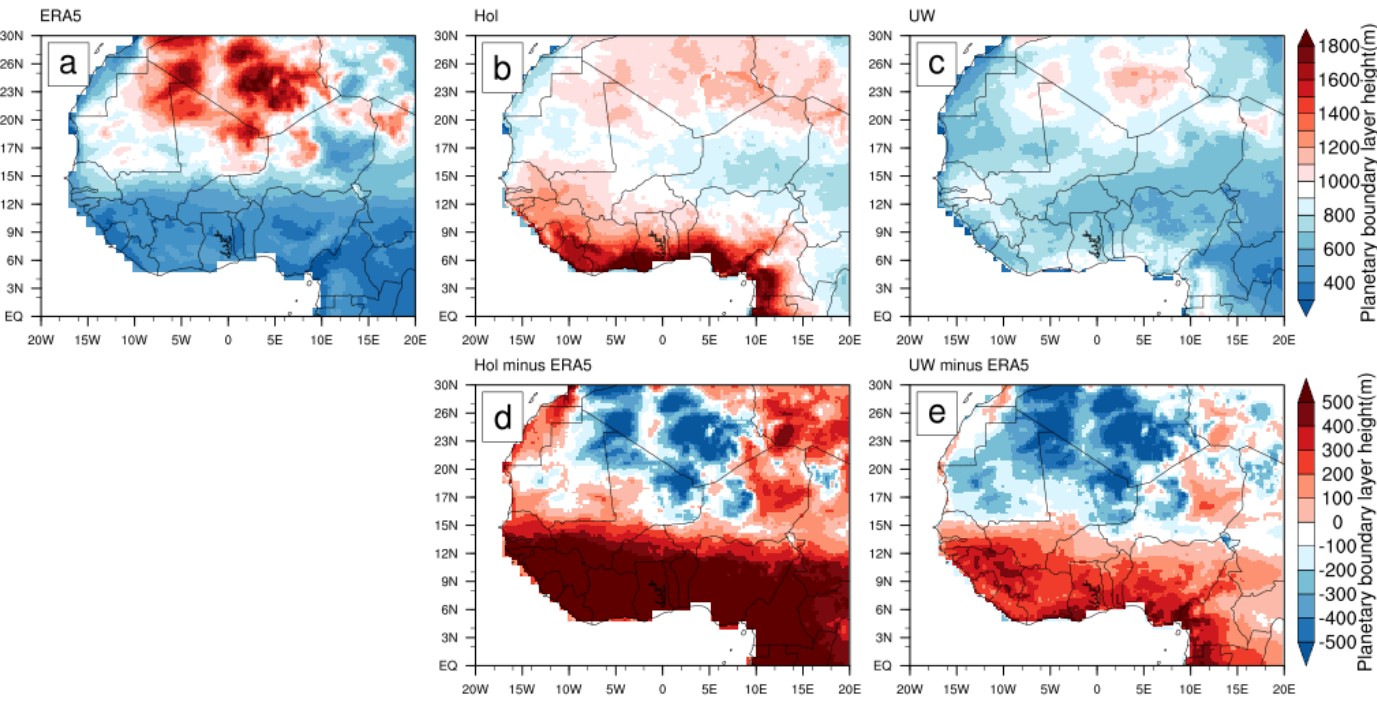

**Figure 10.** Mean summer (JJAS) planetary boundary layer height over West Africa: (**a**) ERA5, (**b**) Hol, and (**c**) UW and their different biased with respect to ERA5 (**d,e**).

The level of SAL is between 800 and 500 hPa [73] and is located around 17° N when the wind moves westward through Sahel towards Atlantic Ocean. Since Flamant et al. [74], SAL events are known to affect ABL in West Africa. The ERA5 distribution of BLH is similar to one found in Ndao et al. [75] study with ERA-Interim. They found that the dry region of SAL corresponds to high BLH values over West Africa from February to October and low values along the Guinean coast until the end of the monsoon front (around 15°) during the summer period (June to September). Hence, in the rainy season, the cool air from the ocean decreases the BLH in the coastal region in a opposite way to the dry air from Sahel, which is characterized by high values of BLH.

Hol failed to capture the patterns observed by ERA5 by displaying, conversely to ERA5, high BLH values along the Guinean coast. UW reproduced the spatial distribution observed relatively well by the ERA5 BLH, with PCC values above 0.7. High values were

observed over the SAL and low values along the Guinean coast, including orographic regions. However, UW generally underestimated the BLH values of ERA5, with a negative mean bias value of approximately 5.

Figure 11 presents the annual cycle of the monthly planetary boundary layer averaged over West Africa and the subregions Guinea Coast and West and Central Sahel, which were central for ERA5 and models Hol and UW. Apart from the Guinean coast region, where Hol failed, in all other subregions and in the entire West African region, both models qualitatively reproduced the seasonal variations observed by ERA5. Nevertheless, with respect to ERA5, Hol presented coarsely overestimated magnitudes, whereas UW reproduced the magnitudes fairly well. For instance, in Central Sahel, while ERA5 and UW showed a common May peak between 600 m and 900 m, Hol located it above 1200 m. The highest magnitudes were observed in the Sahel region (Central and West), and the lowest in Guinea Coast.

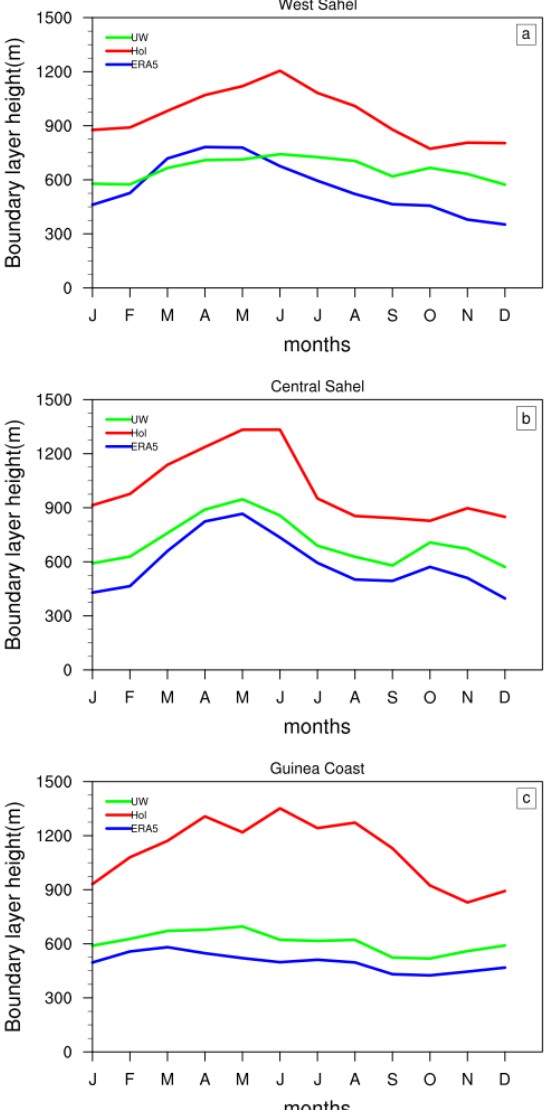

**Figure 11.** Annual cycle of monthly planetary boundary layer height averaged over the subregions Guinea Coast and West and Central Sahel for ERA5 and models Hol and UW. (**a**) West Sahel, (**b**) Central Sahel and (**c**) Guinea Coast.

This result is in line with the findings of Chan and al. [76], who also found that over continents in the subtropics and tropics, PBL depth is typically greater over arid regions and lower over regions with abundant surface moisture. In summary, UW clearly

reproduces both the phases and magnitudes of the annual variation in the BLH. According to Wood [77], BLH is an important determinant of cloud type and, therefore, cloud cover. The seasonal march of the BLH observed here over West Africa (Figure 11) appears to be in phase with the seasonal cloud cover cycle (Figure 7).

This result is consistent with the finding by Wood [78] that over cold subtropical oceans, cloud cover increases as PBL depth decreases as the inversion of cover at the top of the PBL acts as a constraint on vertical moisture transport, maintaining high relative humidity in the PBL and encouraging clouds.

### 3.5. Vertical Profile

Figure 12 shows the mean summer relative humidity and wind speed vertical profile biases of models Hol and UW with respect to ERA5 over (a) West Sahel, (b) Central Sahel, and (d) Guinea Coast.

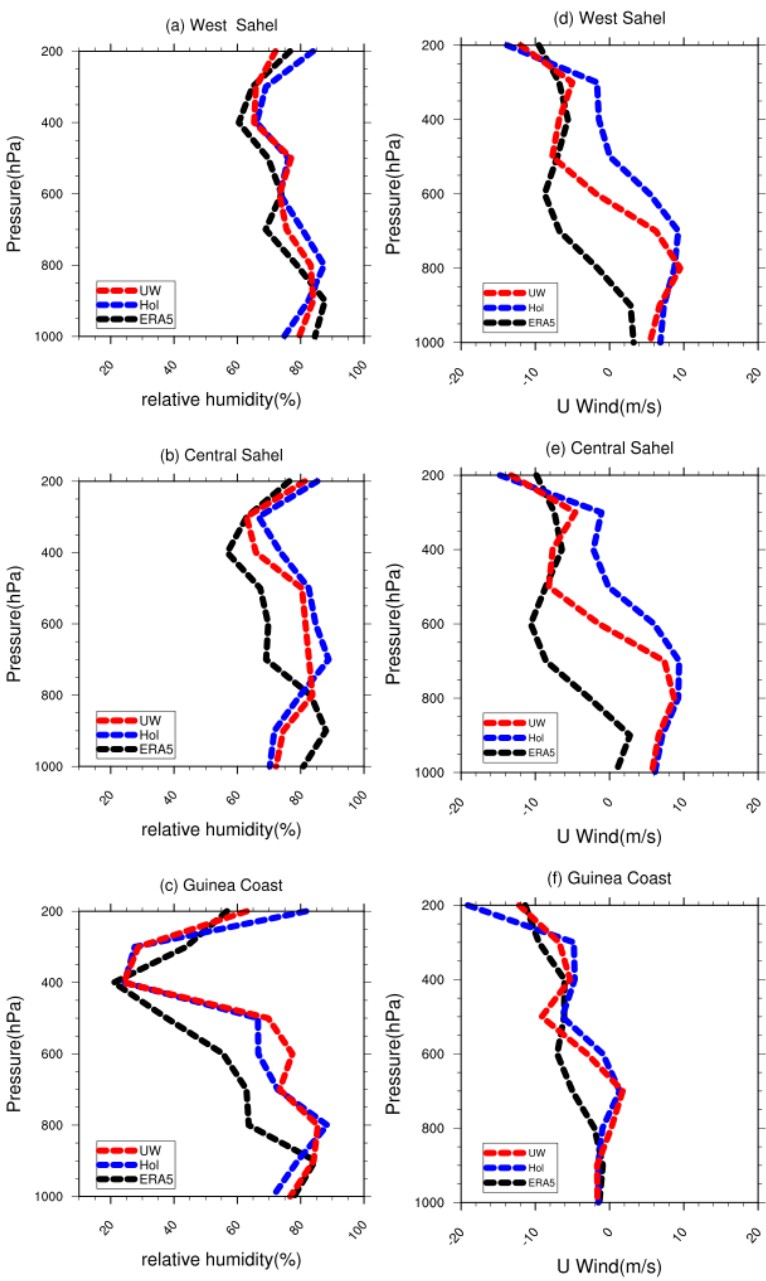

**Figure 12.** Mean summer relative humidity and U-wind vertical profile of ERA5 and models Hol and UW over (**a**,**d**) West Sahel, (**b**,**e**) Central Sahel, and (**c**,**f**) Guinea Coast.

The relative humidity biases of both models in relation to ERA5 have virtually the same vertical structure and relatively similar magnitudes in all three subregions.

The wind speed biases of both models Hol and UW also share the same structure in each subregion and the magnitudes in Guinea Coast and in the lower layers of the Sahelian regions. In West and Central Sahel, between 700 hpa and 300 hpa, both models, although in phase with ERA5, displayed different wind intensities. Unlike Hol, which significantly underestimated the wind speed in this layer compared with ERA5, UW showed wind speeds that are relatively close. Both models seemed to reproduce the phases of the vertical profiles (wind and relative humidity) reasonably well. Nevertheless, UW showed a clear improvement over Hol concerning the intensities of these vertical profiles.

## 4. Conclusions

This study analyzed two different parameterizations of the planetary boundary layer (PBL), namely Holtslag and UW, in the RegCM5 model to verify their performance in simulating the West African climate.

During the JJAS rainy season, both models overestimated total rainfall in the orographic regions. The UW experiment represented total rainfall fairly well compared to its counterpart Hol. Both models reproduced convective rainfall well, with a relatively weak dry bias limited to the Guinean coast subregion.

Generally, statistical analysis of the two models showed that UW yielded better results than Hol for simulating rainfall.

The simulation of near-surface temperature using the two schemes was also considered. The temperature in both models was well reproduced, although the biases were more significant in Hol than in UW. More specifically, the UW scheme was characterized by a cooling effect due to the reduced diffusivity of eddy heat in the lower troposphere, which occurred in this scheme.

The UW scheme succeeded in reproducing the PBL height distribution but underestimated it compared with ERA5, whereas the Hol scheme clearly failed to capture the distribution. Analysis of the annual cycle of PBL across subregions showed that low heights occur during the rainy season but are also localized in the Guinea Coast region. Both Hol and UW reproduced these characteristics well, but compared with ERA5, the heights were much higher in Hol than in UW. The analysis of the simulated total cloud cover can explain the better performance of the UW PBL scheme than the Holtslag scheme in reproducing surface temperature and PBL height. The strong overestimation of PBL heights in Hol was found to be associated with a strong underestimation of total cloud cover.

Overall, the present study shows that both models reproduced most of the climatological characteristics of the West African region. Nevertheless, the differentiation between the two schemes is clearly along Guinea Coast and in orographic regions. In these topographically complex regions, UW appears to be more appropriate than Hol because it displays better results.

A considerable increase in the horizontal resolution of the model could better resolve the PBL, convection, and topography features of the region. Consequently, it could make the difference between the two parameterizations of PBL Hol and UW even more significant.

**Author Contributions:** F.S., conducted the research under the supervision of A.D. (Adama Diawara) and B.K. Methodology and original version preparation was carried out by F.S., A.D. (Adama Diawara), B.K., A.D. (Arona Diedhiou) and A.A.K. participated in the formal analysis, while B.K.K., F.Y., A.B., K.K., D.T.T., A.L.M.Y. and D.I.K. contributed to the discussion of the results and drafting of the manuscript. A.M.L.F. helped with data processing using NCL software. All authors have read and agreed to the published version of the manuscript.

**Funding:** This research was supported by the Institut de Recherche pour le Développement (IRD), France (UMR IGE Imputation, grant no. 252RA5).

**Data Availability Statement:** Data from these simulations is shared free of charge on request by e-mail to konebra75@gmail.com. The ERA5 reanalysis (accessed on 15 September 2023) is available for

download at the electronic address: https://cds.climate.copernicus.eu/cdsapp#!/dataset/reanalysis-era5-single-levels?tab=form (accessed on 15 May 2017) [58]. Global Precipitation Climatology Project version 1.3 (GPCP V1.3) is available for download at the electronic address: https://www.ncei.noaa.gov/data/global-precipitation-climatology-project-gpcp-daily/access/ (accessed on 15 May 2017) [53].

**Acknowledgments:** We would like to thank the Centre national de calcul de Côte d'Ivoire (CNCCI) of the Université Félix Houphouet Boigny (UFHB) for providing the computing power used to carry out the simulations in this work. The authors also thank the Institute of Research for Development (IRD, France) and the Laboratoire Mixte International NEXUS (LMI-NEXUS) (Abidjan, Côte d'Ivoire). The authors are grateful to all students, technicians, engineers, and researchers at ICTP (Abdus Salam International Centre of Theoretical Physics; Trieste, Italy) involved in the development and improvement of the regional climate model RegCM.

**Conflicts of Interest:** The authors declare no potential conflicts of interest.

**Code Availability:** The RegCM5 model open-source code is available for download at https://github.com/ictp-esp/RegCM (accessed on 15 June 2023). The NCL language was used for the computations and the plots, and the source can be found at https://www.ncl.ucar.edu/ (accessed on 15 May 2017).

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
