# Peer review of "Assessment of the Sensitivity of the Mean Climate Simulation over West Africa to Planetary Boundary Layer Parameterization Using RegCM5 Regional Climate Model"

_atmosphere, doi:10.3390/atmos15030332_

Round 1

Reviewer 1 Report

Comments and Suggestions for Authors

Comments on the Quality of English Language

The spelling and grammar need careful revisiting.

Author Response

The authors acknowledge the valuable comments of the reviewers, which helped to improve the manuscript. Updates are highlighted in yellow in the revised version.

Reviewer 2 Report

Comments and Suggestions for Authors

The reviewed manuscript evaluates the sensitivity of the main climate characteristics simulated by RegCM5 to the choice of the PBL parametrization scheme over West Africa. 

The manuscript is generally well constructed. Results are presented in a clean and easy to follow way. However, I do have a feeling that the manuscript was written in a hurry, therefore I recommend to the authors to go thoroughly through the text and systematically correct it. 

Major comments:

1.     Is there a point to use a non-hydrostatical dynamic on a 25km resolution?

2.     Is bilinear interpolation the best choice for precipitation? 

3.     Title should be modified to “Assessment of the Sensitivity of the Mean Climate Simulation over West Africa to Planetary Boundary Layer Parameterization Using RegCM5 Regional Climate Model” or “Sensitivity of the RegCM5 simulated mean climate characteristics to planetary boundary layer parameterization over West Africa” or something similar.

Minor comments:

Line 70: Please check language.

Line 85: This sentence should be in the same paragraph as the previous one.

Line 105: impacts the wind speed and convective precipitation.

Figure 1 – in the results it is constantly referred to the region of complex topography. However, this cannot be seen in this figure. It would be much easier to follow if there is a real topography map or this map with model’s topography is re-plotted with different altitude thresholds in the color bar.

Line 183-185: Please rewrite the sentence.

Lines 239-243: This sentence is too long and difficult to understand. Please rewrite. 

Line 244-249: Please state explicitly which fields were evaluated using ERA5 (t2m, total cloud cover, boundary layer height,…).

Line 279: It should be “their” instead of “they”. But I suggest more detailed figure caption, describing what is shown in each map. Also, units for numbers of color bars are needed. BIAS could be shown as relative number, rather than absolute (Hol-GPCP)/GPCP*100%

Line 350: Hovmoller

Line 351: Degrees symbols are written wrongly.

Line 373: Fig. 3.d

Line 423: There is no mean BIAS shown in the Table 3. RMSE is spelled wrongly as RMSD

Line 426-426: Table 3 caption is copied from the table 2. Please write new caption that describes correctly what is presented in table 3.

Line 430: ERA5(i-j)

Line 443-445: The sentence is not clear, please rewrite. Figure should be probably Fig5-j.

Line 464: degrees symbol. Also, this sentence should be rewritten. 

Line 482: “we” should be left out.

Line 536: Boundary Layer Height

Comments on the Quality of English Language

English is of average quality, and it requires editing by a native speaker.

Author Response

(The authors gave the same response as above.)

Reviewer 3 Report

Comments and Suggestions for Authors

This study in detail investigates the suitable PBL scheme for Western Africa. This is a useful work, but the limitations of the work need to be discussed more clearly. In particular, your evaluations are based on an average of three years, and this means that year-to-year variations and/or temporal variations are not investigated. Another important point to be discussed is that ERA5 is used as reference data for your output variables other than the precipitation. Though ERA5 can be a good reference data, it is a reanalysis data and is not an observation itself. Thus, a possible issue to only use ERA5 as reference data needs to be mentioned. Minor comments are listed below.

Line 96

“Top” should be “TOP” to be consistent with the term that was used in the previous paragraph.

2.1 Model Description.

Please describe the time step for the numerical integration.

Line 165.

There are continued two “uses” in one sentence, and one of them needs to be removed.

Figure 1.

Please add the unit to the color scale.

Line 192.

“inferior” might be “initial”?

Table 1.

“Lateral Boundary Condition” is not included in this table. Please also check the consistency of the word (initial? Inferior?) between this table and the body of the paper.

Figure 2.

Please add the unit to the color scale or explain the units in the caption.

Figure 3.

I think the color scale label in Figure 3(f) is Precipitation (mm/day), not Temperature (°C).

From Line 457 to Line 463 can be combined into one paragraph. It seems that several other paragraphs could be combined into one paragraph for better readability. Please check and consider if some of your independent paragraphs can be combined into one paragraph.

Line 593.

If I look at Figure 12, “700hPa and 300hPa” seems to be more appropriate than “600hPa and 300hPa”.

Line 625.

It seems that the connection between this sentence and the previous paragraph is not clear because you mentioned the inferior performance of the Hol scheme. Thus, I recommend rewording or removing this sentence. Rewording/removing this sentence needs to be also reflected in the corresponding part of your abstract.

Author Response

(The authors gave the same response as above.)
